# Perioperative Care in Colorectal Cancer Surgery before a Structured Implementation Program of the ERAS Protocol in a Regional Network. The Piemonte EASY-NET Project

**DOI:** 10.3390/healthcare10010072

**Published:** 2021-12-31

**Authors:** Luca Pellegrino, Eva Pagano, Marco Ettore Allaix, Mario Morino, Andrea Muratore, Paolo Massucco, Federica Rinaldi, Giovannino Ciccone, Felice Borghi

**Affiliations:** 1Oncological Surgery, Candiolo Cancer Institute-FPO-IRCCS, Candiolo, 10060 Torino, Italy; luca.pellegrino@ircc.it (L.P.); felice.borghi@ircc.it (F.B.); 2Clinical Epidemiology Unit, Città della Salute e della Scienza di Torino, University of Turin and CPO Piemonte, 10126 Torino, Italy; gianni.ciccone@cpo.it; 3Department of Surgical Sciences, Città della Salute e della Scienza di Torino, University of Turin, 10100 Torino, Italy; marcoettore.allaix@unito.it (M.E.A.); mario.morino@unito.it (M.M.); 4Surgical Department, E. Agnelli Hospital, 10064 Pinerolo, Italy; amuratore@aslto3.piemonte.it; 5Department of General and Oncological Surgery, Azienda Ospedaliera Ordine Mauriziano, 10100 Torino, Italy; pmassucco@mauriziano.it; 6School of Medicine, University of Turin, 10100 Turin, Italy; federica.rinaldi@edu.unito.it

**Keywords:** colorectal cancer surgery, ERAS protocol, compliance indicators

## Abstract

Background: In 2019, the Enhanced Recovery After Surgery (ERAS) protocol for colorectal cancer surgery was adopted by a minority of hospitals in Piemonte (4.3 million inhabitants, north-west Italy). The present analysis aims to compare the level of application of the ERAS protocol between hospitals already adopting it (ERAS, N = 3) with the rest of the regional hospitals (non-ERAS, N = 28) and to identify possible obstacles to its application. Methods: All patients surgically treated for a newly diagnosed colorectal cancer during September–November 2019, representing the baseline period of a randomized controlled trial with a cluster stepped-wedge design, were included. Indicators of compliance to the ERAS items were calculated overall and for groups of items (preoperative, intraoperative and postoperative) and analyzed with a multilevel linear model adjusting for patients’ characteristics, considering centers as random effects. Results: Overall, the average level of compliance to the ERAS protocol was 56% among non-ERAS centers (N = 364 patients) and 80% among ERAS ones (N = 79), with a difference of 24% (95% CI: −41.4; −7.3, *p* = 0.0053). For both groups of centers, the lowest level of compliance was recorded for postoperative items (42% and 66%). Sex, age, presence of comorbidities and American Society of Anesthesiologists (ASA) score were not associated with a different probability of compliance to the ERAS protocol. Conclusions: Several items of the ERAS protocol were poorly adopted in colorectal surgery units in the Piemonte region in the baseline period of the ERAS Colon-Rectum Piemonte study and in the ERAS group. No relevant obstacles to the ERAS protocol implementation were identified at patient level.

## 1. Introduction

The Enhanced Recovery After Surgery (ERAS) program is an evidence-based multidisciplinary care pathway which aims to reduce surgical stress and lead to a faster rehabilitation of patients. The ERAS Society has promoted the development of this pathway firstly through the drafting of guidelines in colorectal surgery and then adapting them to different surgical disciplines and procedures [1,2]. Careful adoption of the ERAS program has been shown to reduce length of hospital stay, postoperative complications and costs when compared to traditional care. Despite the advantages, especially in colorectal surgery, the diffusion of the ERAS protocol is still rather low and limited to selected centers. Worldwide, there are only a few experiences of implementation of the ERAS pathway on a regional scale. In Italy, so far, the compliance to this pathway has been on a voluntary basis and heterogeneous.

In the hospital network of the Piemonte region (4.3 million inhabitants, north-west Italy) the average length of stay (LOS) for scheduled interventions was 10 days in 2018, while the corresponding figure in the only ERAS certified hospital was 6 days. Moreover, the average proportion of procedures performed with mini-invasive techniques (laparoscopy or robot assisted) was 66%, with high heterogeneity between centers, in comparison to 82% in the ERAS certified center. On the basis of the regional data and available literature, the systematic adoption of the ERAS protocol was identified as a useful approach to reduce the length of stay and standardize patterns of care in the perioperative period. To support the diffusion of ERAS principles in the hospital network and to estimate its impact, a cluster randomized trial, supported by an audit and feedback strategy, was launched in late 2019.

The first quarter of the ERAS Colon-Rectum Piemonte study evaluated the “standard of care” across all centers previous to the audit and feedback strategy adopted to implement the protocol. The present analysis aims to describe the level of application of the ERAS items for colorectal cancer surgery in a real-world setting—a regional network of hospitals—before the implementation of the standardized protocol. The second aim is to identify obstacles to the ERAS application from the experience of the regional centers before the beginning of the study. Healthcare outcomes are not included at this stage, but they will be analyzed at the end of the trial including all enrolled participants.

## 2. Materials and Methods

The ERAS Colon-Rectum Piemonte study included all hospitals in the Piemonte region expected to perform more than 30 colorectal cancer surgical procedures per year. A pragmatic, cluster, stepped-wedge, randomized trial, with a sample size of more than 2200 patients, was designed with the aim of evaluating the impact in terms of process and outcomes of a regional application of the ERAS protocol in colorectal cancer surgery. Written informed consent was obtained from patients to take part in the study. The study was approved by the ethics committees of the promoting and participating centers. Details on study protocol have been previously published [3].

Data describing perioperative ERAS items, based on the guidelines of the ERAS Society in colorectal surgery, were collected prospectively for each patient.

The present analysis included all patients surgically treated for colorectal cancer during the 3 months first period of the study (September–November 2019), the baseline quarter with standard care.

At baseline, few centers were classified as “ERAS centers” if they had previously received specific training in the ERAS pathway, had a multidisciplinary ERAS team and a protocol approved and applied in their unit. These centers were included in the study as a reference group and not randomized. All other centers were classified as “non-ERAS centers”. During the baseline period all the centers were required to continue with their usual perioperative care and to complete a case report form (CRF) for all the enrolled patients.

Patients’ characteristics between groups of hospitals were compared through the t-test for continuous variables and the chi-square test for categorical variables.

Compliance of each center to ERAS guidelines was measured as the mean percentage of compliance of their patients with a list of indicators (reported in Appendix A, together with rationale and calculation methods). Indicators were calculated overall and for groups of items identified according to the phase of care (preoperative, intraoperative and postoperative).

According to Kehlet’s 2018 proposal [4], compliance to a core set of key items (preoperative counselling, epidural anesthesia in laparotomic, fluid normovolemia, no nasogastric tube and combined with early oral feeding and mobilization) was also assessed.

Compliance to single ERAS items were described graphically, by type of center (ERAS and non-ERAS) and by means of radar charts.

To estimate the difference in compliance levels between ERAS and non-ERAS groups, a multilevel linear model was estimated, adjusting for patients’ characteristics and considering the enrolment centers as random effects. The patients’ covariates included in the models were: sex, age classes (<65, 65–74, ≥75), Charlson Comorbidity Index (0 or ≥1), cancer site (colon or rectum) and ASA score (1–2 or 3–4).

A multilevel regression model with the same set of covariates and the centers as random effects was also applied to estimate the effect of patient demographic and clinical characteristics on the level of compliance to the ERAS items. The model was also stratified by the type of centers group (non-ERAS and ERAS), in order to highlight the possible change in effect due to the presence of an adopted ERAS institutional protocol.

## 3. Results

During the baseline period of the study, 443 enrolled patients in 28 hospitals received colorectal cancer surgery, 364 of whom (82%) were in 25 centers not applying ERAS before the beginning of the study.

Patients’ characteristics are described in Table 1. In non-ERAS centers, patients were slightly older, with a higher presence of females and a higher frequency of comorbidities, but a lower ASA score. In the ERAS centers, operative time was longer and neo-adjuvant therapies were more frequently supplied.

Overall, the level of compliance to the ERAS protocol was 56% among non-ERAS centers and 80% among ERAS ones, with an absolute difference of 24 points between the two groups (95% CI: −41.4; −7.3, *p* = 0.0053) (Table 2). Postoperative items had the lowest level of compliance (42% in the non-ERAS centers versus 66% in the ERAS centers). The largest difference between the two groups was in the preoperative items (27 points, 95% CI: −44.2; −10.7, *p* = 0.0014), where the ERAS group had the highest compliance (91%). Intraoperative items appeared to also be frequently applied among non-ERAS centers (64%) and the difference with ERAS ones was of 16.5 points (95% CI: −31.6; −1.5, *p* = 0.0317).

In the core set of key items, the estimated difference between the two groups was −37 points (95% CI: −59.8; −13.2; *p* = 0.0022), with a compliance of 41.3% in the non-ERAS centers and 78% in the ERAS centers.

Figure 1 describes the level of compliance to single ERAS items, using a radar chart to compare the two groups of centers (non-ERAS and ERAS). Among preoperative items, the non-ERAS group had compliance below 30% for anesthesiological visit time, counselling and carbohydrate loading. Anemia correction and nutritional risk assessment showed 50% and 40% compliance, respectively, whereas all the other items were above 80%. Among ERAS centers, anemia correction was the only preoperative item showing a low level of compliance (46%).

Among intraoperative items, epidural anesthesia in open surgery had a low level of compliance in both groups (44% in non-ERAS and 36% in ERAS), as well as the avoidance of abdominal drainage for colon surgery (44% in non-ERAS and 62% in ERAS). Fluid normovolemia also had low compliance in the non-ERAS group (45%).

In the postoperative phase of care, non-ERAS centers showed low levels of compliance for all the postoperative items but early nasogastric tube removal (65%). Among ERAS centers, the levels of compliance were high overall, except for early removal of intravenous infusion (IV) (47%), early mobilization at 1 day postoperative (56%) and early follow up (60%).

Table 3 shows the results of the models estimating the effect of several covariates on the level of overall compliance to the ERAS protocol; sex, age, presence of comorbidities and ASA score were not associated with different levels of compliance to the ERAS protocol. For rectum cancer cases, however, the level of compliance with all the ERAS items was lower than 4.15 points (95% CI: −6.27; −2.04; *p* = 0.0001) compared to colon cancer cases. These results were confirmed for both groups of centers by a stratified analysis as shown in Appendix A.

## 4. Discussion

This study reports the baseline evaluation of a more complex trial aiming to implement the ERAS pathway throughout all surgical units treating colorectal cancer in a north-western Italian region (Piemonte). As most of the literature is dedicated to describing the implementation of the ERAS protocol in referral centers with a high commitment on its application, we were interested in describing ERAS knowledge and dissemination in a real-world contest before the implementation of a controlled protocol supported by an audit and feedback approach to understand how different the clinical practice from the ERAS standards is.

Despite evidence supporting the ERAS program in colorectal surgery [5], the present analysis shows that several items of the ERAS protocol were poorly adopted in our regional units in 2019. Only 3 centers out of 28 had already introduced a formal and structured ERAS protocol in their daily practice. As expected, the level of compliance to the ERAS items in the non-ERAS group was significantly lower across the entire pathway and the postoperative items were the ones with the lowest level of compliance. Such a low level of compliance was also present for the key items suggested by Kehlet 2018 [4].

Previous surveys [6,7,8] investigating the spontaneous diffusion of the ERAS protocol have demonstrated a low adoption of several items among both surgeons and anesthesiologists, showing a substantial variation in perioperative practice all around the world. A recent Canadian study [9] has shown that a formalized ERAS protocol significantly increases the rates of compliance to intra and postoperative items, such as early diet advancement on postoperative day 0, restrictive use of intravenous fluids, and early catheter and drainage removal. At the same time, some items such as perioperative epidural use and narcotic reduction, balanced postoperative analgesia and early mobilization were similarly widespread in ERAS and non-ERAS centers. The low level of compliance to postoperative items is in line with previous findings of a similar multihospital study in Canada [10].

Our analysis confirms that some ERAS strategies, especially in the pre and intraoperative periods, were already adopted outside of a formal ERAS protocol. This evidence, together with the low compliance to some ERAS items observed in the ERAS group, confirms the need for a structured implementation program to achieve a more complete and homogeneous adoption of the ERAS protocol among the regional centers.

One of the main obstacles to the diffusion of the ERAS pathway was due to the complexity of its global structure and the need for multi-professional and multidisciplinary teams. To overcome these issues some centers have developed “ad hoc”, simplified pathways, including only a few ERAS items considered of major importance. Although a simplified pathway may seem easier to apply and has been suggested as a possible initial focus [11], the application of the entire protocol is also supported by the available evidence. Several reports, including the study from the ERAS Society registry [12], suggest that a reduction in compliance to the ERAS protocol negatively affects short term results such as length of hospital stay, postoperative morbidity and readmission rate. Moreover, a “dose–response” association between the ERAS items’ compliance and patient outcomes improvement is described [13]. Furthermore, a recent Spanish multicenter prospective study [14] showed that high levels of compliance to the ERAS items were correlated with lower rates of complications, infections and mortality. The Spanish study included 2084 consecutive patients scheduled for elective colorectal surgery, divided between those treated or not in a self-declared ERAS center.

As expected, our study documented a significant difference in the total overall compliance with the ERAS items between ERAS (80%) and non-ERAS (56%) centers, indicating a wide margin for improvement.

Multivariate models showed that age, comorbidity, tumor site and ASA score did not appear to be limiting factors to ERAS compliance, while the main obstacles to the diffusion of the protocol itself were identified in the organization of the activity and clinicians’ preconceptions.

The wide differences observed at baseline between the few ERAS centers and the rest of the regional hospitals may be accountable, at least in part, to pre-existing selection factors and not only to the early adoption of the ERAS protocol. However, the true differences at baseline could have been reduced by changes implemented during the discussion of the study protocol and due to the data collection with CRFs that included all the ERAS items, thus mitigating the real pre-existing differences among groups.

In any case, it is expected that this experience will contribute to reducing the regional variability in the surgical management of colorectal cancer with a general improvement of the standard of care. In addition, thanks to the large sample size and the availability of data on all the ERAS components, we will analyze the contribution of each item and its interaction on patient outcomes, with the aim of proposing a more simplified and flexible protocol without jeopardizing its effectiveness.

The final results of this pragmatic trial centers will contribute to reducing the gap of evidence regarding the real applicability and effectiveness of the ERAS when offered to an entire hospital network.

## 5. Conclusions

Despite the established efficacy of the ERAS approach and the general acceptance in the colorectal surgical community, compliance with the ERAS in a regional network of hospitals in the north-west of Italy was found to be low in 2019. Compliance with some items has been shown to be suboptimal even in the centers identified as formally implementing the ERAS protocol. Such a limited level of implementation was not associated with patients’ characteristics, such as presence of comorbidities, high ASA score or older age. Potential obstacles to a full ERAS adoption are likely to be due to organisational and cultural factors. To promote the routine implementation of the ERAS protocol, a structured implementation program supported by an audit and feedback approach could be effective and needs to be assessed.

## Figures and Tables

**Figure 1 healthcare-10-00072-f001:**
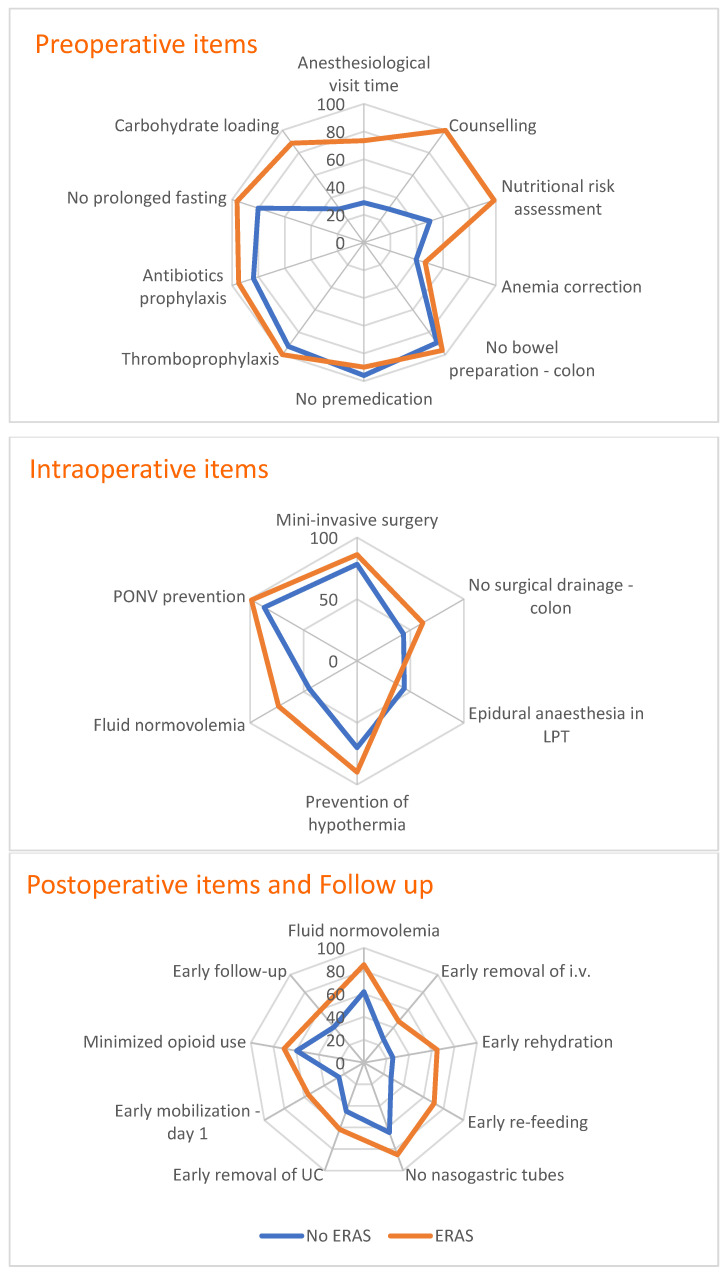
Level of compliance to single ERAS items by type of center (non-ERAS and ERAS). LPT = laparotomic; i.v. = intra venous; UC = urine catheter.

**Table 1 healthcare-10-00072-t001:** Patients’ characteristics at baseline by type of center (non-ERAS and ERAS).

Variables	Non-ERAS (N = 364)		ERAS (N = 79)		*p* Value
	N	%	N	%	
Sex					
Male	210	57.7	55	69.6	0.050
Female	154	42.3	24	30.4	
Age classes					
<65	93	25.5	21	26.6	0.063
65–74	104	28.6	32	40.5	
≥75	167	45.9	26	32.9	
Charlson Comorbidity Index					
Charlson I. = 0	172	47.3	40	50.6	0.586
Charlson I. ≥ 1	192	52.7	39	49.4	
ASA score					
1–2	208	57.1	38	48.1	0.136
3–4	155	42.6	41	51.9	
Unknown	1	0.3			
Cancer location					
Colon	255	70.1	52	65.8	0.460
Rectum	109	29.9	27	34.2	
Neoadjuvant therapy					
Not executed	308	84.6	57	72.2	0.008
Executed	56	15.4	22	27.8	
Type of procedure					
Right colectomy	129	35.4	24	30.4	0.548
Left colectomy	74	20.3	17	21.5	
Transverse colectomy	16	4.4	5	6.3	
PME	30	8.2	4	5.1	
TME	62	17	20	25.3	
Miles’ resection	16	4.4	2	2.5	
Altro	37	10.2	7	8.9	
Stomia					0.599
Present	91	25.0	22	27.8	
Absent	273	75.0	57	72.2	
Type of surgery					
Laparotomy	79	21.7	11	13.9	0.296
Laparoscopy	242	66.5	58	73.4	
Robotic	43	11.8	10	12.7	
Operative time (mean, sd)	224.6	90.5	266.3	116.1	0.003

PME: partial mesorectum excision; TME: total mesorectal excision.

**Table 2 healthcare-10-00072-t002:** Level of compliance (%) to groups of ERAS items by type of center (non-ERAS and ERAS).

ERAS Items	Type of Center						Difference between Non-ERAS and ERAS Centers			
	Non-ERAS (N = 364)			ERAS (N = 79)						
	%	95% CI		%	95% CI		Estimate *	95% CI		*p* Value
Preoperative items	62.7	60.9	64.4	91.3	89.2	93.4	−27.4	−44.2	−10.7	0.001
Intraoperative items	64.4	61.9	66.9	80.7	76.6	84.8	−16.5	−31.6	−1.5	0.032
Postoperative items and follow up	42.4	39.1	45.8	66.2	59.8	72.6	−25.8	−52.2	0.7	0.056
All items	55.7	53.9	57.5	80.0	77.2	82.7	−24.3	−41.4	−7.3	0.005
Core set of key items **	41.3	38.6	44.1	78.0	73.4	82.5	−36.5	−59.8	−13.2	0.002

* Estimated by a multilevel regression model adjusted for patient demographic and clinical characteristics with centers as random effects. ** Preoperative counselling, epidural anesthesia in laparotomic, fluid normovolemia, no nasogastric tube and combined with early oral feeding and mobilization [4].

**Table 3 healthcare-10-00072-t003:** Level of compliance (%) to all the ERAS items by patient demographic and clinical characteristics.

Variables		%	Estimate *	95% CI		*p* Value
Sex	Female	59.72	ref.			
	Male	60.56	−0.88	−2.86	1.10	0.385
Age groups	<65	60.17	ref.			
	65–74	60.28	−2.35	−5.03	0.33	0.086
	≥75	59.84	−1.75	−4.39	0.89	0.193
Cancer site	Colon	60.47	ref.			
	Rectum	59.13	−4.15	−6.27	−2.04	0.0001
Charlson index	=0	58.98	ref.			
	≥1	61.04	1.57	−0.71	3.85	0.176
ASA score	1–2	59.28	ref.			
	3–4	61.03	0.56	−1.77	2.89	0.637

* Estimated by a multilevel regression model adjusted for patient demographics and clinical characteristics with centers as random effects.

## Data Availability

The data that support the findings of this study are available from the corresponding author, E.P., upon reasonable request.

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
