# Peer review of "Perioperative Care in Colorectal Cancer Surgery before a Structured Implementation Program of the ERAS Protocol in a Regional Network. The Piemonte EASY-NET Project"

_healthcare, 2021, doi:10.3390/healthcare10010072_

Round 1

Reviewer 1 Report

The article is well presented, although it speaks of a recurring theme such as the difficulty of implementing the ERAS protocols, in north Italy as in the rest of the world. I would have liked the authors to include some imaginative strategies (differents to the already known), to help hospitals of their region to adopt these protocols. This point would have made the study more interesting.

Author Response

Reviewer 1

The article is well presented, although it speaks of a recurring theme such as the difficulty of implementing the ERAS protocols, in north Italy as in the rest of the world. I would have liked the authors to include some imaginative strategies (differents to the already known), to help hospitals of their region to adopt these protocols. This point would have made the study more interesting.

Reply: The aim of the study was to assess how extensively ERAS items have already been adopted in a real-world context before the implementation of a standardized protocol supported by an Audit&Feedback approach, which is recommended as one of the most effective implementation strategies. As three centres were already implementing ERAS, we have also compared the two groups to assess how complete was the implementation also in this second group of centres. The assessment of the efficacy of the study based on A&F in strengthening the adoption of ERAS at regional level will be the object of the analysis of the complete study.

We have clarified better this choice at the beginning of the Discussion (Pag.5, row 164).

Reviewer 2 Report

Interesting study, which confirms the expected difference in the "real-world" between ERAS and non-ERAS centres. There is quite a lot of variability in "ERAS" units...especially around the use of epidural anaesthesia and MBP. It would be interesting for the authors to produce a table simply looking at Kehlet's 5 most important ERAS components (preoperative patient information, thoracic epidural anesthesia in open (but not laparoscopic) colonic surgery, avoidance of fluid overload and hypovolemia, no nasogastric tube and combined with early oral feeding and mobilization) (Ann Surg 2018:267 (6):998-9).

There are some minor English language issues and it may be worth having this reviewed.

Eg in the Conclusion....Despite the long- and well-established history of the ERAS approach for colorectal cancer surgery and the high level of its knowledge among professionals, its compliance in a regional network of hospitals in the North-West of Italy was find to be low in 2019. .....Suggested text...Despite the established efficacy of the ERAS approach and the general acceptance in the colorectal surgical community, compliance with ERAS in a regional network of hospitals in the northwest of Italy was found to be low in 2019.............Compliance to some items has been shown to be suboptimal even in the centres identified as formally implementing ERAS protocol. Such limited level of implementation was not associated to ......with..... patient’s’ ......patients'......characteristics, like..... such as....presence of comorbidities, high ASA score or older age. Potential obstacles to a full ERAS adoption are likely to be due to organisational and cultural factors. To promote a routinely .....the routine......implementation of the ERAS protocol a structured implementation program supported by an Audit&Feedback approach could be ef-fective and needs to be assessed.

(Suggestion is to change text in red to test in green)

Author Response

Reviewer 2

Interesting study, which confirms the expected difference in the "real-world" between ERAS and non-ERAS centres. There is quite a lot of variability in "ERAS" units...especially around the use of epidural anaesthesia and MBP. It would be interesting for the authors to produce a table simply looking at Kehlet's 5 most important ERAS components (preoperative patient information, thoracic epidural anesthesia in open (but not laparoscopic) colonic surgery, avoidance of fluid overload and hypovolemia, no nasogastric tube and combined with early oral feeding and mobilization) (Ann Surg 2018:267 (6):998-9).

Reply: Thanks for this valuable suggestion. We added the analysis of compliance to the core set of key items [preoperative counselling, Epidural anaesthesia in laparotomic, fluid normovolemia, no nasogastric tube and combined with early oral feeding and mobilization] of the ERAS protocol proposed by Kehlets in 2018.

There are some minor English language issues and it may be worth having this reviewed.

Eg in the Conclusion....Despite the long- and well-established history of the ERAS approach for colorectal cancer surgery and the high level of its knowledge among professionals, its compliance in a regional network of hospitals in the North-West of Italy was find to be low in 2019. .....Suggested text...Despite the established efficacy of the ERAS approach and the general acceptance in the colorectal surgical community, compliance with ERAS in a regional network of hospitals in the northwest of Italy was found to be low in 2019.............Compliance to some items has been shown to be suboptimal even in the centres identified as formally implementing ERAS protocol. Such limited level of implementation was not associated to ......with..... patient’s’ ......patients'......characteristics, like..... such as....presence of comorbidities, high ASA score or older age. Potential obstacles to a full ERAS adoption are likely to be due to organisational and cultural factors. To promote a routinely .....the routine......implementation of the ERAS protocol a structured implementation program supported by an Audit&Feedback approach could be ef-fective and needs to be assessed.

(Suggestion is to change text in red to test in green)

Reply: Thanks for these suggestions. We have revised English language following your suggestions and throughout the manuscript.

Reviewer 3 Report

It's a nice work, well planned and well presented. Good to identify the reasons of a poor implementation of the ERAS protocol, although the scientific value of the this research is average - low: based on how important is this knowledge and how this is going to help the community in the future. I see in the title "perioperative care before the ERAS implemnentation", but my understanding is that the data come after the implementation of the ERAS in ERAS centers and non ERAS centers.

Author Response

Reviewer 3

It's a nice work, well planned and well presented. Good to identify the reasons of a poor implementation of the ERAS protocol, although the scientific value of the this research is average - low: based on how important is this knowledge and how this is going to help the community in the future. I see in the title "perioperative care before the ERAS implemnentation", but my understanding is that the data come after the implementation of the ERAS in ERAS centers and non ERAS centers.

Reply: Data come after the implementation of the ERAS only in three ERAS centres, declaring to be ERAS compliant at the beginning of the study. All the other centres were not applying ERAS in that moment. They were enrolled in the study to support them in the ERAS implementation by means of a controlled protocol supported by an A&F approach. The present analysis refers to the baseline data when routine clinical practice was still adopted by the non-ERAS centres.

We tried to clarify better the aims and the interest of the present results at the beginning of the Discussion (Pag.5, row 164).

Reviewer 4 Report

General comments

This study provides useful information on the adoption of ERAS for colorectal cancer surgery in the Piemonte region. However, there are some serious problems with publishing this research paper in Healthcare.

Specific comments

1) It is expected that there will be regional differences in whether ERAS has been adopted in the hospital or not. What are the implications of the results of this paper for readers outside of the Piemonte region? The authors should clarify novelty and clinical usefulness of this paper.

2) The original purpose of ERAS is to reduce the length of hospital stays, postoperative complications and costs, as mentioned by the authors. Namely, the results and discussion would not be complete without describing the outcome of length of hospital stays, postoperative complications and costs between hospitals with ERAS and those without. If there is no difference in these outcomes, then there would be no need to require compliance with ERAS in the first place.

3) In the Results section, the surgical procedures should be described in detail. Is there any difference in intraoperative blood loss or operative time between the institutions?

Some sigmoid colon tumors may be performed low anterior resection, similar to rectal cancer. Ascending and transverse colon cancers have very different surgical techniques. Suture failure may not be a major problem in patients with colostomy.

These considerations should not be ignored in ERAS study, where postoperative complications are an important factor.

Author Response

Reviewer 4

General comments

This study provides useful information on the adoption of ERAS for colorectal cancer surgery in the Piemonte region. However, there are some serious problems with publishing this research paper in Healthcare.

Specific comments

1) It is expected that there will be regional differences in whether ERAS has been adopted in the hospital or not. What are the implications of the results of this paper for readers outside of the Piemonte region? The authors should clarify novelty and clinical usefulness of this paper.

Reply: As most of the literature is focused on describing the level of adherence to ERAS items in referral centres with a high commitment on its application, we consider of potential to describe ERAS knowledge and dissemination in a real-world contest prior the implementation of a controlled protocol supported by an A&F approach, to understand how different is the clinical practice from ERAS standards. This information, representing an entire regional hospital network, has a general interest and, as far as we know, is scarcely present in the literature. We have better clarified this choice at the beginning of the Discussion (Pag.5, row 164).

2) The original purpose of ERAS is to reduce the length of hospital stays, postoperative complications and costs, as mentioned by the authors. Namely, the results and discussion would not be complete without describing the outcome of length of hospital stays, postoperative complications and costs between hospitals with ERAS and those without. If there is no difference in these outcomes, then there would be no need to require compliance with ERAS in the first place.

Reply: The reviewer comment is certainly relevant. However, as we stated in the last paragraph of the Introduction and in the Discussion, the evaluation of ERAS implementation on healthcare outcomes is the objective of the trial. We have clarified that these data will be analysed, according to the study protocol, on the complete dataset, at the end of the study. The aim of the present analysis is to provide an initial overview on how extensively ERAS items are implemented in a real-world context in absence of specific implementation strategies.

3) In the Results section, the surgical procedures should be described in detail. Is there any difference in intraoperative blood loss or operative time between the institutions?

Some sigmoid colon tumors may be performed low anterior resection, similar to rectal cancer. Ascending and transverse colon cancers have very different surgical techniques. Suture failure may not be a major problem in patients with colostomy.

These considerations should not be ignored in ERAS study, where postoperative complications are an important factor.

Reply: We thank the reviewer for these relevant suggestions. We have therefore modified table 1, adding data on the type of surgical procedure, neoadiuvant therapies, performed stomas and operative time in the two groups of centres.

Round 2

Reviewer 4 Report

The authors well replied to the comments.

However, the scientific value of this research is low because it has no data about the outcome of length of hospital stays, postoperative complications and costs between hospitals with ERAS and those without.

Hospitals without ERAS may already be satisfied with these outcomes and may not be attracted to the idea of implementing ERAS.

The goal of ERAS implementation is not to comply with the protocol of ERAS, but to bring about these outcomes.

Author Response

We completely agree with the reviewer that the matter of the question is the improvement of outcomes and not only the compliance to the ERAS protocol.. Anyway, the need for ERAS implementation in the entire network of regional centres was identified as a priority due to the background situation before the study beginning. In 2018 average length of stay was around 10 days against 6 days in the study coordinator centre (a certified ERAS centre), evidencing the existence of wide margins of improvement in reducing length of stay. The present analysis has the aim to describe the real-world level of application of ERAS, being the impact on clinical outcomes  in relation to ERAS application the main objective of the trial. Probably, we were not clear enough on the background situation bringing to the study implementation. We have added the following sentence in the Introduction, with the hope of improving the clarity of the rationale of the present analysis.

“In the hospital network of Piemonte region (4.3 million inhabitants, North-West Italy) the average length of stay (LOS) for scheduled interventions was 10 days in 2018, while the corresponding figure in the only ERAS certified hospital was 6 days. Moreover, the average proportion of procedures performed with mini-invasive techniques (laparoscopy or robot assisted) was 66%, with high heterogeneity between centres, in comparison to 82% in the ERAS certified centre., On the basis of the regional data and available literature, the systematic adoption of the ERAS protocol was identified as a useful approach to reduce length of stay and standardise patterns of care in the perioperative period. To support the diffusion of ERAS principles in the hospital network and to estimate its impact, a cluster randomized trial, supported by an Audit & Feedback strategy, has been launched in the late 2019”.